# Quality of Life and Psychological Distress Related to Fertility and Pregnancy in AYAs Treated for Gynecological Cancer: A Systematic Review

**DOI:** 10.3390/cancers16203456

**Published:** 2024-10-12

**Authors:** Yaël Stroeken, Florine Hendriks, Jogchum Beltman, Moniek ter Kuile

**Affiliations:** Department of Obstetrics and Gynecology, Leiden University Medical Center, 2333 ZA Leiden, The Netherlands; f.j.hendriks@lumc.nl (F.H.); j.j.beltman@lumc.nl (J.B.); m.m.ter_kuile@lumc.nl (M.t.K.)

**Keywords:** Adolescents and Young Adults, gynecologic oncology, fertility preservation, fertility-sparing surgery, Quality of Life, Psychological outcomes

## Abstract

**Simple Summary:**

For Adolescents and Young Adults (AYAs) with gynecological cancer, preserving fertility is crucial for their Quality of Life. This systematic review explores how possible (in)fertility affects treatment choice and psychological well-being during and after treatment. The studies included were evaluated using the Mixed Methods Appraisal Tool (MMAT). A total of 15 studies were analyzed. The key findings emphasize the importance of informing all AYAs about how cancer treatment can impact fertility and discussing available fertility preservation options. This review also highlights that AYAs with gynecological cancer who do not have fertility preservation options are often overlooked. Therefore, research should focus more on this specific group. We recommend improving the Shared Decision-Making (SDM) and follow-up process for AYAs with gynecological cancer by addressing fertility-related questions and concerns, which could enhance their long-term Quality of Life.

**Abstract:**

Background/Objectives: With growing survival rates for Adolescent and Young Adults (AYAs) diagnosed with gynecological cancer, the focus shifted to Quality of Life (QoL). Fertility-sparing surgery offers a viable alternative to standard, usually fertility-impairing treatments. Treatment choice remains difficult and renders perspectives of AYAs on decision-making and psychological outcomes afterwards. This review examines the impact of (in)fertility on psychological well-being both during cancer treatment, and in the long term. Methods: A systematic review of the peer-reviewed literature was conducted by searching Pubmed, Web of Science, Cochrane Trial database and PsycINFO on 30 November 2023. The review included studies with a focus on gynecological cancer, fertility and pregnancy related psychological outcomes, QoL, and psychosocial factors influencing decision-making. Case reports and reviews were excluded. Quality was assessed with the Mixed Methods Appraisal Tool (MMAT). Results: 15 studies, published between 2005 and 2023, involving 1328 participants, were included. Key findings highlight the significance of informing all AYAs about cancer treatment effects on fertility and discussing fertility preservation options. Feeling time-pressured and conflicted between choosing the best oncological outcomes and preserving fertility were common. Factors such as younger age at diagnosis, time pressure, and inadequate counseling by healthcare workers increased reproductive concerns which contributed to long term psychological distress. Research on AYAs with gynecological cancer without fertility preservation possibilities is limited and should be prioritized. Conclusions: This review shows that both Shared Decision-Making (SDM) and follow-up processes can be improved by addressing fertility-related questions and concerns, therefore increasing long-term QoL. This review is registered in PROSPERO (ID 448119).

## 1. Introduction

Annually, approximately 160,000 female Adolescents and Young Adults (AYAs), age 15 to 39 years old, ref. [1] are diagnosed with gynecological cancer globally [2]. In high-income countries, more than 75% of these individuals survive their disease [2]. Consequently, focus has shifted to improving the Quality of Life (QoL). Preserving fertility and the ability to conceive are important factors influencing the QoL among AYAs [3,4]. Research shows that almost 75% percent of these AYAs express a desire to conceive in the future [5]. The standard treatment for cervical, endometrial and ovarian cancer involves surgery on internal genitalia, which directly impairs the fertility and future pregnancy potential of patients. For that reason, alternative treatment options have been developed in the last few decades. For many AYAs with early-stage (mainly FIGO stage I disease) gynecological cancer, Fertility-Sparing Surgery (FSS) (in combination with chemotherapy or hormonal therapy), preserving the uterus and at least one ovary, has become an option [6]. 

In general, the QoL for AYAs with cancer is lower compared to their healthy peers, with ‘fertility concerns’ as a significant factor influencing QoL [7]. Given that the average age for having a first child in Europe is currently 29.7 years old and continues to rise [8], many AYAs may not have considered having children yet or might just be in the process of doing so. Additionally, since the peak incidence for example of cervical cancer diagnosis in AYAs occurs between the ages of 30 and 34 [9], many AYAs are forced to delay their plans for pregnancy due to cancer treatment and follow-up care. Research suggests that the possibility of preserving fertility and becoming pregnant in the future can help AYAs better cope with cancer [10,11]. Adequate fertility counseling has proven to reduce decisional conflict and long-term treatment-related regret in women with breast and gynecological cancer [5,10,12]. However, many studies have focused on breast and childhood cancer survivors, and there is limited research exploring the evidence on psychological burden in AYAs with gynecological cancer. Although FSS targets preserving fertility, it still has a major effect on the function of internal genitalia, which differs significantly from chemotherapy and fertility preservation (FP) methods for breast and childhood cancer survivors. It is therefore crucial to consider that women with gynecological cancer may have different perspectives on fertility and pursuing pregnancy. 

Previous studies on AYAs with gynecological cancer mainly focused on oncologic outcomes, improving counseling for FP and FSS, pregnancy rates, and sexual functioning [10,13,14].

This is the first systematic review examining the impact of cancer treatment on QoL and psychological burden in relation to fertility and pregnancy for AYAs with gynecological cancer. Our aim is to enhance a better understanding of how FSS and non-FSS affect QoL related to fertility and pregnancy of women with gynecological cancer exclusively. This will help healthcare professionals to provide more informed and personalized guidance to these patients who are considering FSS or FP. 

## 2. Materials and Methods

### 2.1. Search Strategy and Study Selection

This systematic search was conducted in accordance with the Preferred Reporting Items for Systematic reviews and Meta-Analyses (PRISMA) guidelines [15]. 

The following electronic databases were searched: PubMed library, Web of Science, PsycINFO and Cochrane trial library on the 30 of November 2023. Search terms contained: gynecological cancer, treatment options, fertility- and pregnancy related psychological outcomes, QoL, and psychosocial factors. (Full searches: see Appendix A).

### 2.2. Eligibility Criteria and Paper Selection

The inclusion criteria were peer-reviewed English original articles, quantitative, qualitative or mixed methods design, addressing patient reported outcome, and experiences and concerns related to fertility and pregnancy, of AYAs treated for gynecological cancer. Reviews and case reports were excluded. In case multiple diagnoses were studied, the article was included when: 1. A clear subgroup analysis for AYAs with gynecological cancer was performed; or 2. non-significant differences between studies groups were clearly provided; or 3. AYAs with gynecological cancer were the largest sample, with a minimum of 20%.

The literature search was independently performed by two researchers (YS and FH) and developed together with experienced librarians (JP) and (EvG). The search results were exported to Endnote 20. Based on title and abstract, a first article selection was made by YS and FH. Then full-text reading was conducted and carefully interpreted for the final selection. Any discordance on study selection was resolved by consensus. If no agreement was obtained, a third observer (MtK) was consulted to meet consensus. Additionally, to ensure the most complete search, reference lists of other reviews were manually searched.

The Mixed Methods Appraisal Tool (MMAT) [16], a widely used instrument that acts as a quality appraisal of quantitative, qualitative and mixed method studies, was independently used by YS and FH to grade the quality of the studies. If no agreement was obtained, the opinion of MtK was consulted to gain consensus.

Our review is registered in PROSPERO (ID 448119). 

## 3. Results

In total, 3549 articles were collected in Endnote. First, 1104 duplicates were removed. After reviewing the titles and abstracts of the remaining articles, it was found that 2397 articles did not align with the scope of this review. After excluding these, 48 articles were selected for full-text analysis. Another 33 articles were excluded for not meeting the inclusion criteria. One article [17] was included by screening the reference lists of other reviews. One article [18] did not meet the requirements of the MMAT screening questions. In total, fifteen articles met the inclusion criteria and were selected for this systematic review. Two studies (by Wenzel et al.) analyzed the same cohort. In one study, AYAs with gynecological cancer were compared to other cancer survivors [19], and in one study, they were compared to healthy controls [20]. These papers were combined when describing the results, reducing the total to fourteen studies. Furthermore, two studies (by Carter et al.), contained preliminary data in one article [21] which was expanded in the second [22]. The first study contained qualitative data, so these studies were counted separately [21], but the absolute number of AYAs with gynecological cancer was counted as one. Details of the study selection process are shown in the PRISMA flowchart, Figure 1.

### 3.1. Overview of Studies 

#### 3.1.1. Study Characteristics

The studies were published between 2005 and 2023. Eight quantitative studies focused on psychological outcomes and QoL related to fertility and pregnancy [20,23,24,25,26,27,28,29], four qualitative studies explored attitudes and fertility and pregnancy-related psychosocial outcomes [17,21,30,31], and two mixed method studies [22,32] were included with varying designs: retrospective, cross-sectional, and prospective cohort, using questionnaires, interviews and focus groups. Most studies were conducted in the USA *n* = 7 and Europe *n* = 3. The remaining studies were conducted in Australia (*n* = 1), China (*n* = 1), Japan (*n* = 1) and Saudi Arabia (*n* = 1).

#### 3.1.2. Patient Characteristics

In total, 1328 AYAs with gynecological cancer participated. Most studies consisted completely of AYAs with gynecological cancer (9/14), while in others (5/14), AYAs with gynecological cancer were a smaller part of the group, varying between 8 and 88%. Within the total sample (*n* = 1328), the subdivision of different cancer types was: cervical cancer *n* = 744 (56%), ovarian cancer *n* = 316 (24%), endometrial cancer *n* = 196 (15%), and other types *n* = 15 (1%) like vaginal, vulvar, gestational trophoblastic, and pelvic cancer. One study did not define subtypes (*n* = 57) [29]. 

The age range of the AYAs varied between 15–46 years old. Most studies reported a mean age of 32 or older, with the majority of participants being over 18 years old [25]. The time between diagnosis and study period varied from diagnosis [21] to eleven years post-diagnosis [24]. 

Although all papers studied attitudes, psychological factors, and QoL related to fertility and pregnancy of AYAs with gynecological cancer, half of the studies (*n* = 7) lacked relevant information on FSS indication, and type of FP or FSS treatment. In four studies, information on counseling for fertility options or FSS eligibility [22,27,28,30] was missing, and in three others, it was also unknown whether FSS was received [19,20,23].

Furthermore, information on parity at diagnosis was available in only five studies [17,22,24,29,31], and only seven studies provided information on childbearing or desire for more children at survey [17,23,24,25,27,31,32]. 

All fourteen studies examined baseline characteristics, time since diagnosis, and decision-making regarding FSS and FP. Some studies compared outcomes to women with other cancer types [27,28,29], women with infertility unrelated to cancer, ref. [28] or healthy subjects [19,20]. Most studies also explored the impact of reproductive concerns on psychological well-being and QoL. Furthermore, attitudes from relatives and doctors, and cultural expectations, were examined [17,27,30]. 

#### 3.1.3. Quality Assessment

Fourteen articles were graded according to MMAT criteria [33] (see Appendix A). Most studies were of good or reasonable quality. Sample sizes of interview studies varied from 12 or 14 participants [17,30], raising questions about saturation, to 20 or 29, which was reasonable and good [21,31]. In quantitative studies, response rates varied between 30–95%. Most articles did not specify differences between responders and non-responders, and in four articles, the response rate was not described at all [17,22,29,30]. Therefore, response bias cannot be ruled out. An overview of the articles and key findings can be found in Table 1. 

### 3.2. Key Thematic Findings

This systematic review examined the attitudes, psychological factors, and QoL related to fertility and pregnancy in AYAs with gynecological cancer. Recurring themes were categorized into three timeframes: ‘The decision-making process’, ‘Reflecting back on decisions’, and ‘Looking toward to (future) fertility and pregnancy’. We will discuss factors influencing decision-making, decisional regret, psychosocial outcomes associated with fertility (loss) and pregnancy, and QoL post-treatment, as well as AYAs’ attitudes and knowledge on these subjects. 

#### 3.2.1. The Decision-Making Process

##### Attitudes Toward (Future) Fertility and Treatment Choice

Preserving fertility and desiring a ‘future’ pregnancy were key factors for choosing FSS among AYAs [17,19,21,22,24,28,30,31,32]. Five studies examined the attitudes of women toward the decision-making process. A prospective, quantitative cohort study analyzed a group of AYAs with cervical cancer (*n* = 71) choosing either Radical Trachelectomy (RT) or Radical Hysterectomy (RH). These AYAs were followed from time at diagnosis until two years after treatment. Women opting for RT were younger at diagnosis, with a mean of 32 years (RT) vs. 38 years (RH) old, and had no children 84% (RT) vs. 43% (RH). A total of 98% and 46% mentioned that fertility and desire for future children, respectively, influenced their treatment choice [22]. Schlossman et al., studied women with different gynecological tumors (*n* = 55), (mean age at diagnosis 32), 73% considered future children, and 80% received fertility counseling. No association was found between women’s age and FP choice, but AYAs with children at diagnosis were 72% less likely to undergo FP [32]. In another study, 71% (*n* = 36) of AYAs with gynecological cancer (mean age at diagnosis 35) who lost their fertility reported that becoming a parent remained their most important goal. Additionally, 69% of the group felt they lacked time to complete childbearing before treatment. The importance of preserving fertility was emphasized in two retrospective, qualitative studies on women who received FSS for cervical cancer (*n* = 15) and ovarian cancer (*n* = 12). They indicated that the hope for a biological child, maintaining control over their body and reproductive choices, feeling more feminine and complete when sparing the internal genitalia, played a significant role in decision-making for FSS [17,30]. Furthermore, AYAs expressed feeling very anxious after diagnosis about the possible fertility loss [17,30]. Multiple quotes underline this, for example: “I have been more terrified of surviving the chemo and cancer and then losing my fertility, than not surviving at all” [30]. While the importance of fertility preservation was also emphasized in another study (mean age at diagnosis 32), they found that 67% of these women did not choose FSS, or FP techniques such as oocyte preservation (*n* = 37). AYAs indicated common barriers, including treatment delay, feeling emotionally overwhelmed, and inadequate counseling. However, adequate counseling did not lead to an increase in fertility preservation uptake [32].

Two qualitative studies and one quantitative study [26,30,31] explored decisional conflict in relation to treatment choice. In these studies (mean age at diagnosis ranging from 28–34), half of the women reported not having considered children before diagnosis. This caused internal conflict, as they had to balance their best chances of survival with preserving their fertility. This difficult decision-making process was further complicated by time-pressure [26,30,31]. 

In one study, decisional conflict and psychological distress were measured using validated questionnaires immediately after FP counseling in 16 endometrial cancer patients (mean age at diagnosis 34). The results showed that these women had high decisional conflict scores, indicating significant struggles during decision-making. Furthermore, higher levels of decisional conflict were associated with moderate to severe scores for depression and anxiety [26]. 

Decision-making and perspectives on fertility and disease are also influenced by social and cultural norms. This is evident in three qualitative studies [17,30,31] (mean age at diagnosis ranging from 28–33). Partners’ and family members’ opinions were generally viewed as valuable and supportive [31]. In some cases, however, women expressed that family opinions were powerful and considerably impacted their final choice which caused internal conflict [17]. Additionally, doctors’ opinions were also influential, often favoring the most radical procedure with the highest survival changes, thereby prioritizing FP as a secondary consideration. Several women described feeling pressured by opinions from others, underscoring the impact on their final decision [30]. 

#### 3.2.2. Reflecting Back on Decisions Regarding FSS and FP

##### Decisional Regret

Most studies on QoL and psychosocial outcomes related to treatment and fertility used retrospective designs and examined thoughts on decisions, and factors associated with decisional and treatment-related regret, a few years after treatment [24,26,27,31]. 

In one study, the overall regret score was assessed immediately after the FP decision, and before the commencement of FP procedures. The total regret score was low, indicating mild regret. In this study, all women received fertility counseling from a fertility specialist, followed by FP. As mentioned earlier, these women reported high levels of decisional conflict [26]. One qualitative [24] and one quantitative study [31], investigated factors that could be associated with long-term decisional regret. In a cross-sectional study of 470 gynecological cancer survivors, all eligible for FSS (mean age at diagnosis 34) and on average 11 years after diagnosis, only 46% (*n* = 216) of them recalled fertility counseling, and 39% (*n* = 183) received FSS. Adjusting for age, it was found that recalling, being satisfied with counseling, and receiving FSS were associated with lower regret [24]. Among women all eligible for FSS, there was no association between undergoing FSS and regret. While the majority (69%) had children before treatment, half of them wished to have more children, which was associated with a higher level of regret [24].

In a qualitative study including 14 gynecological cancer survivors treated with FSS (mean age at diagnosis 33), many women stated that the cancer diagnosis had a major impact on their relationship, sometimes even causing break-ups. This was also emphasized in the following quote: “I had a partner at the time but not the same partner as I have now. With the old partner, he was all for the hysterectomy. We’re not together and me and my current partner would like to have children.” Overall, it was stated that women who felt in control during the decision had lower levels of decisional regret afterward [31]. In another study (mean age at diagnosis 34), treatment-related regret was measured with a single item: ‘I regret having undergone treatment that altered my fertility’. This item mediated the association between desire to have children, and fertility-related distress, alongside emotional response to the cancer diagnosis [24]. In two other studies, women expressed frustration with the limited number of support groups for AYAs, which could have helped them coping with cancer diagnosis and fertility-related concerns [23,32].

##### Knowledge and Attitudes Toward (Future) Fertility and Pregnancy

In a cross-sectional study, knowledge about the impact of cancer treatment on conception and offspring, as well as desire to have children was examined [25]. A group of 23 women (mean age at diagnosis 22) who received FSS for ovarian cancer were asked whether they believed that their treatment could affect their ‘reproductive outcomes’ and ‘health of their future offspring’. Nearly half of the women stated that FSS could have damaged their reproductive potential. Only 9% of the women were worried about the impact of FSS on their future offspring, while 52% responded with ‘I don’t know’ [25]. In two studies, approximately half of the AYAs reported that their family planning had not changed due to treatment. In one of these studies, all of the women had received FSS (mean age at diagnosis 22) [25], while in the other, half of the AYAs had received FSS or FP (mean age at diagnosis 32) [32]. It is unclear what percentage of AYAs belonged in both groups. 

Doctors’ attitudes towards childbearing after FSS from patients’ perspective were pointed out in a qualitative study. One woman considered doctors’ comments helpful: “My doctor reminded me of the point by saying, ‘You’d better try hard for half a year or a year because you underwent the surgery for that’. I had almost forgotten it Oh, yes. That’s right.”, but another woman stated that she felt pressured for having children due to perceived expectations from doctors: “To preserve the uterus means to have the ability to conceive. I strongly feel that the hospital people expect me to get pregnant soon and have a child. That direction is better for them.” [17]. 

It is also essential to focus on AYAs who were not eligible for FSS. Unfortunately, this subgroup was not examined in most studies. Only one study investigated the knowledge and attitudes of AYAs regarding third-party options such as Assisted Reproductive Technologies (ART), oocyte donation and surrogacy, and adoption [28]. In this study [28], 51 infertile gynecological cancer patients (mean age at diagnosis 35) were surveyed. The study revealed that 98% of these patients had heard of surrogacy, almost 75% heard of oocyte donation. A total of 50% of participants reported experiencing constraints on their available reproductive options, expressing a lack of consideration for third-party alternatives as part of their family building strategies. However, a detailed exploration of their considerations for these alternatives has not been undertaken [28]. 

#### 3.2.3. Reproductive Concerns and Psychological Distress

##### Baseline Characteristics

Eight of the fourteen studies studied reproductive concerns, seven were cross-sectional [19,20,22,23,25,28,29,30], and one qualitative [17].

The studies found that women expressed at least some reproductive concerns at diagnosis [17,30] or after treatment [19,20,23,25,28,29]. In two studies [22,25], which included only women who received FSS, it was found that 50 to 100% of the women experienced reproductive concerns after diagnosis. 

Several cross-sectional studies [19,20,24,29], conducted between less than 2 up to 11 years after diagnosis, indicated that reproductive concerns were associated with baseline characteristics such as undergoing chemo-or radiotherapy, desire for having children, not having children yet, younger age, and not being in a relationship. This confirms that reproductive concerns are dependent on the stage of life and may change over the course of life. Many women (mean age at diagnosis 35) expressed concerns about time constraints in pursuing childbirth [28]. This was also emphasized in a qualitative study, where women (mean age at diagnosis 28) expressed time pressure after diagnosis, especially when choosing FSS or FP, and during the period after treatment when conception was contraindicated [30]. In a study involving 129 gynecological cancer and 35 breast cancer patients, average time since diagnosis of 3.4 years and mean age at diagnosis 35, researchers investigated factors associated with fertility-related distress. Only 26% of these AYAs received FSS. They found that desire for (more) children was a significant factor associated with fertility-related distress. Not having children or being single did not contribute to fertility-related distress, above and beyond the wish for (more) children. Additionally, the desire for children was significantly higher in AYAs with a Polish background compared to a UK background, indicating that cultural background indirectly affected the level of fertility-related distress [27]. 

Mattsson et al. (mean age at diagnosis 33) investigated psychological distress and its associated factors, in gynecological cancer survivors (*n* = 337), using a cross-sectional design. They found that 85% of the survivors reported psychological distress, with 27% attributing to fertility-related concerns [23]. Additionally, reproductive concerns after treatment, were retrospectively explained as ‘inability to talk openly about fertility’, ‘frustration, grief, guilt and sadness related to infertility’ [19,20,23,28]. 

##### FSS vs. Non-FSS

One study compared AYAs with cancer who received fertility counseling and FP (such as oocyte or embryo banking) with AYAs who received fertility counseling but chose not to undergo FP (19% of the total study population) [29] (mean age of these groups are unknown). The level of reproductive concerns were similar between the two groups. In another study, AYAs with cancer-related infertility who expressed the need for more information on third-party reproductive options showed higher levels of depression and cognitive avoidance (not letting thoughts about the disease in) than survivors who did not require more information [28].

##### Time Since Diagnosis 

It is important to emphasize that reproductive concerns can vary over time, for example before and after cancer treatment and depending on the stage of life. In one study, women were asked before FSS (mean age at diagnosis 33) if they felt confident about conceiving. Before treatment, 62% of the women said they felt confident, and this feeling remained consistent, with 60% of them feeling confident up to two years after surgery. Concerns about getting pregnant and pregnancy itself, decreased from 91% to 73% over the same time [22]. Although many women had concerns about their ability to have children, only 18% of the AYAs had spoken to a fertility specialist in the two-year period after their diagnosis. 

Another cross-sectional study (mean age at diagnosis unknown) found that a shorter time since diagnosis was associated with higher levels of reproductive concerns [29]. However, this study looked at different time frames: less than 2 years, 2 to 5 years and over 5 years since diagnosis. These studies complement each other and offer a broader perspective. 

##### Comparison to Other AYAs with Cancer and Healthy Counterparts

Three studies compared levels of reproductive concerns in AYAs with gynecological cancer with healthy AYAs, AYAs with different types of cancer, or AYAs with infertility without cancer history [19,20,28]. One study compared reproductive concerns of AYAs with gynecological cancer (*n* = 51) to AYAs with infertility without cancer history (*n* = 50), 4–10 years after diagnosis. The reproductive concerns scale (RCS) was found to be elevated in both groups but did not significantly differ [28]. 

Both studies by Wenzel et al. found that cervical cancer survivors (FSS or non-FSS status unspecified, mean age at diagnosis = 33) (*n* = 51), expressed significantly more reproductive concerns compared to their age-matched healthy counterparts. Although AYAs with cervical cancer had higher RCS scores than other cancer patients, no significant difference was found. Overall, higher levels of reproductive concerns are associated with poorer QoL, more cancer-specific and psychological distress, independent of cancer type or fertility status [19,20,28]. Furthermore, they showed that having reproductive concerns was also associated with less social support, lower spiritual well-being scores, more significant gynecologic pain, and poorer sexual functioning [19,20]. 

## 4. Discussion

This systematic review aimed to examine fertility-related distress from the decision-making process until years after treatment in AYAs with gynecological cancer. The study findings suggest that being well-informed about the cancer treatment effects on fertility, being able to discuss fertility-sparing options, and feeling supported by both relatives and healthcare workers in decision-making, were all related to improved long-term psychological well-being. 

The importance of fertility-related counseling and the ability to maintain control for AYAs, both crucial aspects of shared decision-making, were also found in previous studies on AYAs with gynecological and other types of cancer and seem to play a pivotal role in the prevention of long-term psychological distress [5,12,34]. Furthermore, our study shows that several factors, such as young age, desire for children or not having a partner at diagnosis (both dependent on the stage of life) and differences in cultural background also have an effect on long-term psychological distress. Unfortunately, the focus on the shared decision-making process, which also consider the previously mentioned factors, is still insufficiently addressed in both research and clinical practice [5,10]. This could be explained by multiple underlying causes, such as the healthcare providers’ lack of awareness or time, insufficient availability of (written) information, and focus of both patient and doctor on survival [35]. As most AYAs with gynecological cancer and their health care providers tend to focus on treatment options during the counseling process, fueled by the will to survive, fertility concerns, and fertility preservation options seem to be at lower priority [35]. Our study also revealed that many AYAs, when surveyed years later, did not remember receiving fertility counseling at the time of diagnosis. This oversight may be due to recall bias, but since fertility is such an important topic for these AYAs, it is possible that counseling on this topic was not adequately addressed. A consultation, with a gynecologist-oncologist or physician specialized in reproductive medicine, supported by a shared decision-making (SDM) tool on both FP and FSS could help overcome these previous mentioned barriers for both doctors and AYAs with gynecological cancer, and consequently reduce long-term fertility-related distress. Several SDM tools have been developed and proven to be beneficial in counseling for women with cancer, increasing knowledge and lowering decisional conflict [36,37]. In the Netherlands, an SDM tool has been developed specifically for female AYAs with cancer and a wish for FP, including AYAs with gynecological cancer, and is currently being implemented [38]. However, this SDM tool focuses only on FP, and for AYAs with gynecological cancer, the surgical effect of FSS, as well as ethical considerations such as possible spread of disease during FP and delaying cancer treatment, should be integrated. 

In almost all of the studies, an emphasis was put on women with early-stage cancer with the option for FSS. Only a few studies included women who did not have the option for FSS and instead underwent radical surgery or (chemo)radiation therapy [27,28,29,32]. Women who underwent non-FSS and did not recall counseling reported higher levels of regret and distress related to fertility compared to those who recalled counseling [24,31]. Additionally, higher levels of depression were found in infertile women (non-FSS), who perceived the need for more information on third-party fertility options (such as surrogacy or egg donation) after gynecological cancer treatment [28]. As approximately 35 percent of the women with gynecological cancer in high-income countries, such as the Netherlands, are diagnosed in advanced-stage disease and hence FSS is considered no option [39], more attention should be provided to this large group of women in research and in clinical practice. Since this group appears to have higher levels of depression, regret, and distress, it is important to know if these women would have benefited from FP options at diagnosis. We therefore believe that counseling on fertility (and FP) should be provided to all AYAs with gynecological cancer, regardless of their stage of disease or type of treatment, provided that the cancer treatment has curative intent. Dutch guidelines on AYAs with cancer have adopted this type of counseling on fertility and FP [40], but it seems that this has not been widely practiced so far in women with gynecological cancer. Financial and ethical constraints; such as delayed treatment, spread of disease and possible increased risk of recurrence, are also barriers that could hamper further counseling on FP, and third-party options in women who are not able to carry their own child because of treatment [41], depending on the patients’ health insurance and varying regulations on surrogacy. Therefore, additional counseling possibilities on FP often remain context-specific. Nonetheless, the therapy effect of surgery, chemotherapy, or radiotherapy on fertility, and actively asking if the AYA would like a consultation with a fertility specialist without any obligations, should be discussed with all AYAs with gynecological cancer. 

Studies also show that only fifty percent of the AYAs become pregnant after FSS [42]. In the general population, 80% of the women will become pregnant within 6 months of actively trying to conceive [43]. The reasons for the relatively large number of women who do not become pregnant after FSS compared to the general population remain largely unclear. Apart from the treatment effect on fertility, and older age at actively conceiving due to cancer treatment delay this could also be related to underlying causes such as reproductive and pregnancy-related concerns, which seem to be higher in AYAs with (gynecological) cancer compared to the healthy population [19,20]. This could be caused by previously mentioned underlying factors, such as insufficient information on fertility and FP, or economic constraints. In addition to this, our study showed that although many AYAs requested more information on fertility and pregnancy options after treatment, only a few AYAs actually went to see a fertility specialist. This might also be subject to the clinician’s knowledge and attitudes, as a previous study has shown that discussing fertility issues and referring patients to fertility specialists depends on the clinician’s specialty, confidence in knowledge regarding fertility issues and a lack of reproductive specialists in their region. However, specific reasons for these differences remain unclear and should be studied further [44]. In addition to strengthening the shared decision-making process, there should also be a focus on improving follow-up visits to address potential fertility-related questions and concerns once cancer treatment has finished. Previous studies have shown that AYAs are prone to more long-term mental health issues compared to the general population [45]. By doing so, women can be referred to a fertility specialist or other caregivers, such as a psychologist, based on their specific fertility-related concerns, thus potentially improving their long-term Quality of Life. The lack of knowledge regarding the reasons why these women do not (want to) become pregnant or do not visit a fertility specialist after treatment further underscores the significant research gap in this particular subgroup of patients. Currently, a nationwide study known as the PROFIT study, is conducted to investigate on oncological, fertility-and pregnancy outcomes, and long-term QoL of women treated for gynecological cancer. These results are expected to provide a better understanding of underlying reasons for the relatively low pregnancy rates in AYAs, and of long-term psychological distress, such as reproductive concerns. This can ultimately lead to an improved management of AYAs treated for gynecological cancer.

Some limitations should be considered when interpreting the study results. Firstly, it is important to acknowledge that choosing non-FSS as a treatment option is substantially different from undergoing radical treatment without having an actual choice to preserve treatment, as we have pointed out earlier. In many of the included papers, information on the choice for FSS and FP was unknown [19,20,22,23,27,28,30]. In another three articles only a section of the population received fertility counseling and had a choice in treatment, but no subgroup analysis was conducted [24,29,32]. The authors of the articles were approached for additional information, but only one author responded and clarified the missing information [30]. This shows that thorough research on women with infertility after gynecological cancer treatment is still lacking and justifies the need for systematic research, such as the previously mentioned PROFIT-study. Furthermore, the desire for children and the intention to keep fertility options open were crucial factors in decision-making; two-thirds of the studies did not provide information on parity at diagnosis, the perceived number of children at the time of the survey, or the desire for more children at that time. Additionally, most participants were over 32 years old on average, and the median age was often not reported. This presents a significant limitation of these studies, as these baseline characteristics could influence the results. Therefore, caution should be exercised when interpretating these findings. Finally, although we chose to include articles that focused exclusively on AYAs with gynecological cancer or met specific conditions described in the method section to limit heterogeneity, the included studies employed various study designs with both (un)validated questionnaires, interviews, and different outcome measures. This variability complicated the direct comparison of the studies and the generalizability of the results. Furthermore, the percentages of the subgroups; mainly endometrial, cervical and ovarian cancer, do not reflect an accurate distribution across the world. 

## 5. Conclusions

This review highlights the complexity of decision-making on FSS and FP in AYAs with gynecological cancer. It emphasizes the importance of (future) fertility options for AYAs with cancer and the need for shared decision-making on FSS and FP. This should involve an individualized approach that takes patients’ characteristics and preferences into account and should be a part of the standard care for all AYAs with gynecological cancer. 

Additionally, there is a lack of studies on long-term QoL in AYAs without a choice for FP or FSS, and existing data on AYAs who were counseled for FSS or FP is insufficient. Therefore, research on this group should be prioritized. Addressing fertility-related questions and concerns can improve both the shared decision-making and follow-up processes, ultimately enhancing long-term QoL. 

## Figures and Tables

**Figure 1 cancers-16-03456-f001:**
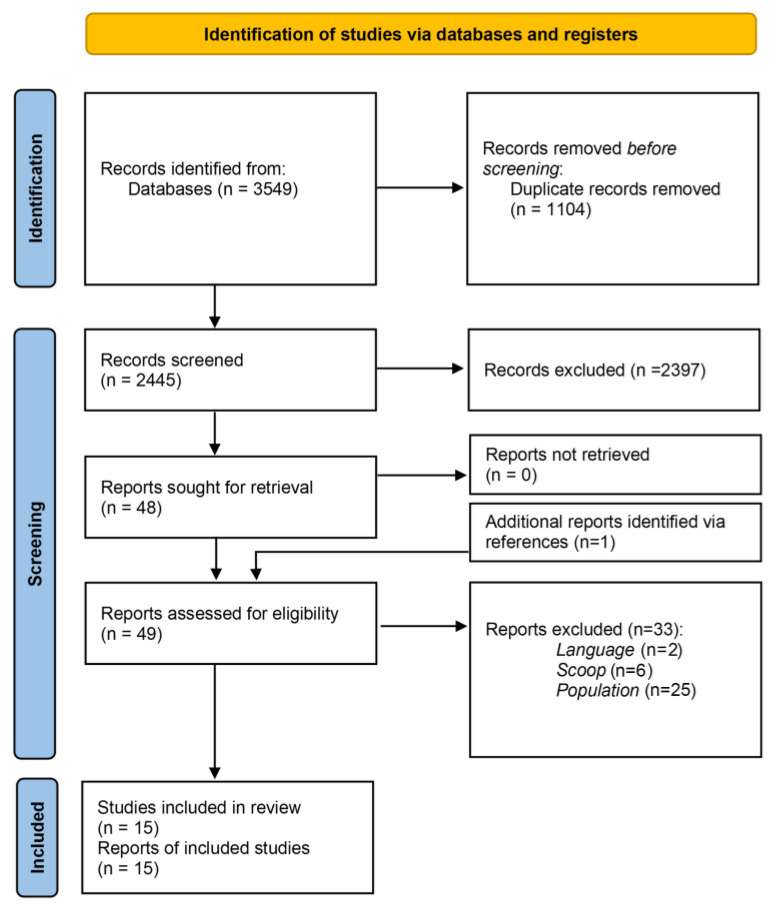
PRISMA flowchart summarizing selection process eligible articles.

**Table 1 cancers-16-03456-t001:** Article characteristics and results.

Study, Origin	Aim	Inclusion Criteria, Recruitment Sites, Study Period	Type of Study, Study Design, Measurements	Sample Size, Age at Diagnosis (Mean and Range), Parity at Diagnosis	Fertility Counseling, FP/FSS	Response Rate, Time Since Diagnosis (Mean)	Relevant Outcomes
Bentsen et al. 2023 Denmark [30]	Exploring thoughts about fertility	AYAs with cancer 18–39 years old at recruitment Recruitment in one hospital 2020–2021	Qualitative Cross-sectional Semi-structured interviews	*n* = 12; Ovarian cancer *n* = 3/12 (25%) age 28 (20–35) parity at diagnosis unknown	Counseling (1/3) (33%), FSS 33%	Unknown Unknown (4 AYAs still received cancer therapy)	Theme identification: • Keeping hope on future children • Participants felt time pressure to choose for fertility preservation and being forced to wait for pregnancy due to treatment • The women faced existential and ethical choices about survival and family formation • Loss of control over the body and treatment
Carter et al. 2007, USA [21]	Exploring reproductive and emotional concerns associated with FSS	Cervical cancer patients 18 and 45 years old at diagnosis, with a pregnancy wish. Planned for radical trachelectomy Recruitment in one hospital 2004–2006	Qualitative Prospective: Pre-operative, after 3, 6, 12, 18, and 24 months post-surgery Reproductive concerns: Self-developed open-ended questions	*n* = 29 age 32 (23–40) parity at diagnosis unknown	Fertility counseling: (100%) FSS: (100%)	Pre-operative 83% 3-months post-surgery 68% 6-months post-surgery 82% *N*/A	• Women stated during the pre-operative assessment that they were concerned about their ability to conceive and maintaining a pregnancy • Women stated they felt time-pressured because of their age, and limited time for conception • Several women reported concerns about the impact of cancer history and treatment on their future baby
Carter et al. 2010, USA [28]	Exploring emotions, reproductive concerns and QoL Determining whether knowledge of, and access to third-party (surrogacy, egg donation) reproduction influences psychosocial outcomes	18–49 years old at recruitment; (1) AYAs with gynecologic cancer (2) BMT/SCT survivors (3) Control group: non cancer patients, wait-list for egg donation (1,2,3) Did not start or has completed childbearing Recruitment in two centers 2006–2009	Quantitative Cross-sectional Reproductive concerns: (RCS) Depression: CES-D Distress: IES (Validated questionnaires) Self-developed questionnaires on perception of third-party parenting, reproductive informational needs, health-related concerns	*n* = 172 GYN-survivors *n* = 51 (30%), age 35 (21–46) BMT/SCT = 71 (41%), age 23 (4–45) non-cancer Infertile control group = 50 (29%) parity at diagnosis unknown	Counseling (% unknown), FP none	Gynecologic 89%, BMT/SCT: 81%, Control group 74% GYN survivors: 4 years and BMT/SCT survivors: 10 years	• No significant differences between mood, reproductive concerns, and mental health QOL were identified between infertile groups (cancer vs. non-cancer) • Cancer survivors (cervical + BMT/SCT) with a perceived need for more information on reproductive options had significantly higher depression and avoidance scores than women reporting no need for more information about reproductive options • Women with infertility without cancer history had significantly higher distress levels than women with gynecological cancer history • 55% of GYN-survivors had the feeling they did not have options for Assisted Reproductive Technique. • 71% of GYN-survivors rated being a parent as most important in their lives
Carter, Sonoda 2010, USA [22]	Exploring emotional functioning, reproductive concerns and QOL	AYAs with cervical cancer with a choice for RT or RH. Recruitment in clinics 2004–2008	Quantitative Prospective Attitudes toward treatment choice and the decision-making process: exploratory items Reproductive concerns: Exploratory items	*n* = 71 RT group *n* = 43 (61%) age 33 RH group *n* = 28 (39%) age 38 nulliparous RT group *n* = 36 (84%), RH group *n* = 12 (43%)	Counseling (% unknown) FSS 61%	Unknown N/A	• 48% of RT vs. 8.6% of RH patients stated they have had adequate time to complete childbearing and stated this was one of the factors for their treatment decision • Women in RT group were younger than women in the RH group (mean age 33 vs. 38) • Fertility was for 98% vs. 46% in RT and RH group one of the most important factors for their treatment decision • Only the RT group was asked about their concerns on: 1. conceiving a baby in the future. Pre-operatively 91% of the AYAs indicated concerns. This percentage decreased to 73% after two years • 2. Feeling successful at conceiving in the future on a scale from 0–100% Pre-operatively, the mean rating was 62% (25–100) decreasing to 55% after 1 year and restoring to 60% after 2 years • After two years 21% was attempting conception, 15% had successfully conceived, and 9% had achieved a successful pregnancy • Only 18% of AYAs had spoken to a fertility specialist within the first two years after surgery
Chan et al. 2017, USA [24]	Determining the effect of FSS on regret, and characterize patients with highest risk of regret	Gynecological cancer patients, 18 to 40 years old at diagnosis Recruitment via a state Registry 2010–2013	Quantitative Cross-sectional Questionnaires: FP (yes/no) Decisional regret: DRS (Validated questionnaire)	*n* = 470, age 34 (18–40) Cervical = 228 (49%) Ovarian = 125 (26%) Endometrial = 117 (25%) nulliparous *n* = 146 (31%)	Fertility counseling (46%) FP/FSS (39%) Self-reported	79% 11 years	• All AYAs were eligible for FSS. Only 46% recalled receiving fertility counseling • After adjusting for age, recalling (satisfactory) counseling and receiving FSS were associated with lower regret • Half of the women desired more children after treatment • Desire for children at time of diagnosis was associated with increased decisional regret (*p* < 0.001)
Chen et al. 2022, China [26]	Exploring emotional state and perceptions on FP/FSS decisions	Diagnosed with gynecological cancer or a benign abnormality on the ovary >18 years old Counseled for FP Recruitment in one hospital 2019–2021	Quantitative Cross-sectional Decisional conflict: DCS Decisional Regret: DRS (Validated questionnaires)	*n* = 33 Female = 16 (48%), age 34 (range unknown) parity at diagnosis unknown	Fertility counseling (100%) FP/FSS (100%)	80% Directly after fertility preservation	• Decisional conflict score on FP/FSS decision was high in women with gynecological cancer • Women felt conflicted between chances on survival and chances on preservation of fertility • Women felt supported by others • DRS-score for decision on FP/FSS was low
Komatsu 2014, Japan [17]	Exploring the experience of AYAs on FP/FSS	AYAs with cervical cancer treated with radical trachelectomy Recruitment in one hospital 2011–2012	Qualitative Cross-sectional, Semi-structured Interview: Perceptions on FP/FSS Perceptions on conception and childbearing after FSS	*n* = 15 age 32 (25–38) nulliparous *n* = 14 (93%)	Fertility counseling (100%) FP/FSS (100%)	Unknown 2 years (median)	• *n* = 3 (20%) had one or more child(ren) after treatment. Themes: • The possibility of the uterus being removed caused a lot of anxiety • FP/FSS repairs the threatened feminine identity in women with cervical cancer • FP/FSS does not simply mean childbearing, maintaining the ability to conceive is significant • Some women imagined their childless life as the worst case scenario
Mattsson et al. 2018, Sweden [23]	Determining the prevalence and predictors of fertility-related distress	Patients with gynecological cancer between 2008–2016 Recruitment from the national cancer registry 2016 -unknown	Quantitative Cross-sectional Cancer-related distress (Non-validated self-developed questionnaire)	*n* = 337 age 33 (range 19–39) Cervical cancer *n* = 224 (66%) Other *n* = 112 (33%) parity at diagnosis unknown	Fertility counseling (% Unknown) FP/FSS (% Unknown)	52% 3 Years	• *n* = 121 (36%) of the women did not have children at time of survey • *n* = 79 (27%) of the women experienced fertility concerns • Cancer related distress was more common in women without children (90%) than with children (82%) *p* = 0.046 • Women stated that support groups with other AYA cancer patients were lacking
Sait 2011, Saudi-Arabia [25]	Exploring emotional attitude on fertility and pregnancy after treatment	AYAs with ovarian cancer treated with FSS between 2000–2010 Recruitment in one hospital Unknown	Quantitative Cross-sectional Phone call survey 3 questions on emotional attitudes extracted from a validated questionnaire	*n* = 23 age 22 (4–35) parity at diagnosis unknown	Fertility counseling (100%) FP/FSS (100%)	59% Unknown	• *n* = 35 (90%) of the women did not have children at time of survey • Almost half of the women stated that the chosen treatment will have an impact on the desire to have children in the future *n* = 10 (44%) • Half of the women were afraid that the received FSS would damage their reproductive potential *n* = 12 (52%) • Few women were concerned that FSS would have an effect on their (future) offspring *n* = 2 (9%)
Schlossman et al. 2023, USA [32]	Exploring barriers of women toward fertility care after treatment and exploring experiences around fertility	Treated for gynecological cancer and still in follow up Recruitment from an academic state hospital 2012–2022	Mixed methods Cross-sectional survey and semi-structured interviews Self-developed questionnaire on the effect of cancer diagnosis and counseling on childbearing and fertility plans	Questionnaire study: *n* = 55 age 32 Cervical *n* = 23 Ovarian *n* = 20 Endometrial *n* = 12 Interview study: *n* = 20 parity at diagnosis unknown, *n* = 26 (47%) had never been pregnant before diagnosis	Fertility counseling (80%) FP/FSS (53%)	24% Unknown	• *n* = 40 (73%) was considering (more) children in the future • Half of the AYAs reported that their family planning goals had not changed due to treatment, one third said it had drastically changed • AYAs who had not undergone fertility preservation stated that this was due to not wanting to delay treatment (49%), feeling to emotionally burdened (18%), or having received inadequate counseling (14%) • There was no statistical difference between FP/FSS and no FP/FSS in terms of age, having a partner at diagnosis and the quality of counseling • Women who had children at time of diagnosis were 72% less likely to undergo FP compared to women who did not have children • During the interviews, 80% of women stated they had not met their family planning goals yet • Women who did not seek fertility care, said this was because of: inadequate counseling (60%), lack of time (60%), economic constraints (55%) and cancer treatment prioritization (55%) • Multiple AYAs were frustrated with the lack of AYA support groups
Sobota and Ozakinci, 2018, United Kingdom, Poland [27]	Determining the associated factors of distress related to reproductive issues and assessing potential impact of socio-cultural background	History of gynecological- or breast cancer Recruitment via multiple outpatient clinics Unknown	Quantitative Cross-sectional survey Psychological wellbeing: IES-R Illness perceptions: Brief-IPQ Value of Children: VOC (Validated questionnaires) Decisional regret: (1 Unvalidated question) Prediction model: Fertility-related distress	*n* = 164 age 35 (19–46) Gynecological = 129 (79%) parity at diagnosis unknown	Fertility counseling (% Unknown) FP/FSS (26%)	47% 3 years	• *n* = 80 (49%) of the women was nulliparous at time of survey • Strongest associated factors for fertility-related distress: Treatment-related regret (*p* < 0.05) Emotional response to cancer diagnosis (*p* < 0.01) Polish background compared to British background (*p* < 0.01) • Desire for more children was correlated with treatment-related regret However, the desire for more children was higher in the group of women with a Polish background and background therefore seen as an indirect effect
Standen et al. 2020, Australia [31]	Exploring attitudes to conception after FSS	Treated for gynecological cancer with FSS between 2005–2016 Recruitment from a tumour board database Unknown	Qualitative Semi-structured phone call interview Theme development: perceptions and knowledge of FSS, and on fertility services	*n* = 14 age 33 (range unknown) Cervical *n* = 6 Endometrial *n* = 3 Ovarian *n* = 5 (and their partners) nulliparous *n* = 7 (50%)	Fertility counseling (100%) FP/FSS (100%)	30% Unknown	• ±50% of the women had not considered children at time of diagnosis and were forced to do so • One woman had conceived a child after treatment while four women attempted IVF • Women who underwent involuntary secondary surgery with complete removal of the uterus or ovaries expressed regret and advised others only to do this if childbearing is completed • Cancer diagnosis and possible infertility had major impact on (future) relationships
Wenzel et al. 2005, USA [19] Wenzel, de Alba 2005, USA [20]	Exploring psychosocial and reproductive concerns, and QoL	Treated for gynecological cancer or lymphoma 5–10 years earlier compared to healthy controls Recruitment from two cancer registry programs Unknown	Quantitative Cross-sectional Phone call-or email survey Quality of Life: QOL-CS Psychological distress: IES Reproductive concerns: Self-developed scale Self-reported infertility	*n* = 379 Cervical *n* = 51 (31%), age 37 (range unknown) GTT *n* = 111 (69%) age 33 (range unknown) age-matched controls = 148 Parity unknown	Fertility counseling (% Unknown) FP/FSS (% Unknown)	Cervical: 88% GTT: 95% Cervical cancer group: 8 years GTT group: 7 years	• AYAs with cervical cancer reported satisfactory QoL • Mental and physical state did not significantly differ from healthy controls • Among cancer survivors, reproductive concerns were associated with lower QoL (*p* < 0.001), poorer physical and mental health • Together with less social support and having physical gynecological problems, reproductive concerns were accounted for 63% of the variance in QoL • More reproductive concerns were reported by AYAs with cancer compared to healthy controls (*p* < 0.0001) • Reproductive concerns were associated with the inability to have children (*p* < 0.001) • Self-reported infertile women had poorer mental health (*p* < 0.05), more cancer-specific distress (*p* < 0.01), and lower overall (*p* < 0.001), physical (*p* < 0.01), and psychological (*p* < 0.001) well-being than women who were fertile
Young et al. 2019, USA [29]	Exploring the association between cancer type and treatment with reproductive concerns	15–35 years old at diagnosis and completed primary treatment Recruitment from national cancer registries 2015–2017	Quantitative Cross-sectional Survey at study baseline and serial online surveys over 18 months Reproductive concerns: RCAC	*n* = 747 Gynecological *n* = 57 (8%) mean age unknown (15–35) nulliparous *n* = 559 (75%)	Fertility counseling (19%) FP/FSS (12%)	Unknown >5 years (64%) 2–5 years (24.5%) <2 years (11.4%)	• No significant differences in reproductive concerns were found between cancer types • 44% of the participants had moderate to high reproductive concerns • Reproductive concerns were similar between the group of women receiving counseling and FP/FSS, and receiving counseling without FP/FSS • Higher reproductive concerns were associated with not having children at diagnosis (*p* < 0.001) and at time of questionnaire (*p* < 0.001), younger age (*p* < 0.001), not having a relationship at time of questionnaire (*p* < 0.001), and lower income (*p* = 0.03)

BMT: Bone Marrow Transplant, CES-D: Center for Epidemiologic Studies Depression Scale, DCS: Decisional Conflict Scale, DRS: Decisional Regret Scale, FP: Fertility Preservation, FSS: Fertility Sparing Surgery, GTD: Gestational Trophoblastic Disease, GTT: Gestational Trophoblastic Tumors, IES-R: Impact of Events Scale-revised, IESL: Interpersonal Support Evaluation List, QoL: Quality of Life, QoL-Cs: Quality of Life-Cancer Specific, RCS: Reproductive Concerns Scale, RCAC: Reproductive Concerns after Cancer Scale, RH: Radical Hysterectomy, RT: Radical Trachelectomy, SCT: Stem Cell Transplant, VOC: Value of Children.

## Data Availability

The raw data supporting the conclusions of this article will be made available by the authors on request.

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
