# Peer review of "Quality of Life and Psychological Distress Related to Fertility and Pregnancy in AYAs Treated for Gynecological Cancer: A Systematic Review"

_cancers, 2024, doi:10.3390/cancers16203456_

Round 1
Reviewer 1 Report
Comments and Suggestions for Authors
Quality of life and survival issue are both important area in treating cancer patients. In addition, fertility is non-negotiable topic in women’s cancer especially in her reproductive period.
I have minor suggestion.
1. What’s boundary, guideline and chance for gynecologic cancer in terms of fertility-Staging I, II, III, IV, Is it impossible among advanced case? Or recurrent cases? Is there fertility sparing radiation indication? How’s uterus transplantation after hysterectomy? Do we adopt more aggressive chemotherapy cystectomy instead of oophorectomy, hysteroscopic resection instead of hysterectomy, wide conization or trachelectomy instead of hysterectomy? How’s future teratogenic effect of conventional chemotherapy and target treatment? How’s bad survival because of delayed treatment due to ovary function preservation?
2. What’s the chance of successful normal pregnancy rate among healthy person compared to GY/other cancer patients? How’s assisted reproductive method for improving pregnancy rate? How much is needed for oocyte preservation and success rate? Do all patients need a psychologic consult for seriousness of fertility & survival?
3. Relatively minor endometrial cancer population (15%) compared to study population of USA (7) and Europe (3). Is there more option for Increasing AYA endometrial cancer in western world? How’s fertility value difference among ethnicity (Asian vs Western) or physician/hospital choice bias difference (More aggressive physician vs non-aggressive, fertility supporting facility affiliated vs non)?
Comments on the Quality of English LanguageNot much
Reviewer 2 Report
Comments and Suggestions for Authors
The authors conducted a systematic review on psychological distress in AYAs with gynecological cancer.
They included 15 manuscripts. Please mention the publication period of these manuscripts also in the abstracts.
The presentation of the results is not at all mature and a Table 1 reaching from page 5-15 with columns that comprise 2 words are simply inacceptable.
Since AYAs can be aged from 15 to 46 years, an information that is painfully missing is how many persons had already children before their cancer diagnosis and how many were nonparous and also how many women became pregnant and gave birth to a healthy child after their cancer treatment.
The views on fertility preservation are obviously different for adolescents and for a woman in her thirties and moreover forties.... This should be taken into account and reported and discussed in all paragraphs
Comments on the Quality of English Languagefor me ok
Round 2
Reviewer 2 Report
Comments and Suggestions for Authors
The quality and the readability have been considerably improved - no further issues for me